# Influence of Experimental Conditions during Synthesis on the Physicochemical Properties of the SPION/Hydroxyapatite Nanocomposite for Magnetic Hyperthermia Application

**Dinh Thi Mai Thanh [1],\*, Nguyen Thu Phuong [2]**  **, Do Thi Hai [3], Ha Ngan Giang [1], Nguyen Thi Thom [2]**, **Pham Thi Nam [2], Nguyen Trung Dung [4]**, **Michael Giersig [5] and Magdalena Osial [5],\***

1 University of Science and Technology of Hanoi, Vietnam Academy of Science and Technology, 18 Hoang Quoc Viet, Cau Giay, Hanoi 10000, Vietnam
2 Institute for Tropical Technology, Vietnam Academy of Science and Technology, 18 Hoang Quoc Viet, Cau Giay, Hanoi 10000, Vietnam
3 Faculty of Basic Science, Hanoi University of Mining and Geology, 18 Pho Vien, Duc Thang, Nam Tu Liem, Hanoi 10000, Vietnam
4 Faculty of Physical and Chemical Engineering, Le Quy Don Technical University, 236 Hoang Quoc Viet, Bac Tu Liem, Hanoi 10000, Vietnam
5 Department of Theory of Continuous Media and Nanostructures, Institute of the Fundamental Technological Research, Polish Academy of Sciences, Pawińskiego 5B Str., 02-106 Warsaw, Poland
\* Correspondence: dinh-thi-mai.thanh@usth.edu.vn (D.T.M.T.); mosial@ippt.pan.pl (M.O.); Tel.: +84-914256885 (D.T.M.T.); +48-22-826-12-81 (ext. 240/463) (M.O.)

**Abstract:** In this work, we report on the fabrication of nanocomposites based on superparamagnetic iron oxide nanoparticles (SPIONs) in combination with hydroxyapatite (HAp) as a platform for drug delivery and magnetic hyperthermia application. First, the influence of experimental conditions such as co-precipitant, bath temperature, and pH on the morphology and magnetic properties of SPIONs was investigated. Then, the superparamagnetic particles were coated with the hydroxyapatite layer for further loading of anticancer drugs, determining the optimal thickness of the HAp shell. The composite was fabricated by the wet chemical process and was characterized by optimizing the experimental conditions of the wiring synthesis to obtain the superparamagnetic spherical material with a high HAp loading as a platform for drug uptake. SEM and TEM studies confirmed the round shape of the magnetic core up to 15 nm in size with a well-defined HAp shell. After checking the material's superparamagnetic properties, the temperature dependence on time and alternating magnetic field strength was tested and optimized in hyperthermia experiments.

**Keywords:** superparamagnetic iron oxide nanoparticles (SPIONs); hydroxyapatite; nanocomposite; magnetic hyperthermia



## 1. Introduction

Cancer is one of the fatal diseases affecting people globally. According to the World Health Organization (WHO), cancer is a leading cause of death worldwide, accounting for nearly 10 million deaths in 2020 [1]. Research and clinical trials are currently focusing on developing new techniques that destroy only cancer cells without affecting the surrounding normal cells. Along with the development of nanotechnology and nanomaterials, superparamagnetic iron oxide nanoparticles (SPIONs) have recently gained interest in cancer theranostic applications such as magnetic resonance imaging (MRI), magnetic hyperthermia, etc., due to their unique features, especially the reaction with magnetic force and their biodegradability [2–5]. SPIONs can be easily synthesized, and thanks to their superparamagnetic properties, SPIONs' suspension can be used as heat generators to defeat cancer cells [6,7].

Since cancer cells are more sensitive to temperature than normal cells, therapeutic temperature change can be induced by the thermogenesis of a high-frequency alternating magnetic field. Based on the small size of these nanoparticles, they can overcome most of the biological barriers and be precisely injected into the tissues. However, magnetic hyperthermia based on SPIONs can induce death for cells when heating up to 41–46 °C for 12 min [8,9]. Moreover, using SPIONs directly is strongly discouraged due to their biofouling in blood plasma, which increases agglomeration and segregation by the reticuloendothelial system [10]. On the other hand, the SPIONs agglomeration would decrease their internal superparamagnetic properties [11]. Therefore, the surface modification of SPIONs is necessary to reduce the agglomeration as well as the biofouling in physiological conditions. The composition and the surface modification of SPIONs play an essential role in therapeutic applications; therefore, the optimization of their physicochemical properties allows cancer treatment [12,13]. There are many approaches to SPIONs modification for their biocompatibility and functionality, such as polymers, ceramics, composites, etc., and polyethylene glycol [14], dextran [15], citrates [16], 3-phosphono propionic acid [17], alginates [18], chitosan [19], polyvinyl alcohol [20], polyvinylpyrrolidone [21], etc., are reported as SPION modification agents. SPIONs can also be easily bioconjugated within the covalent chemistry and/or physical interactions to increase their functionalities, especially the cancer therapy [22] or cancer immunotherapy [23].

Among many compounds proposed as biocompatible also offering the high surface area is hydroxyapatite (HAp), the main inorganic component of human bone and teeth, which has excellent bioactive, biodegradability, and osteoconductivity [24]. It enhances bone growth and osseointegrates, which is applied in dental, orthopedic, and maxillofacial fields [25]. Thus, HAp material is considered a promising candidate for bone substitute material. This biocompatible material can be coated to the surface of nanoparticles to ensure safety for medical applications. HAp has a porous structure; it provides a large surface area, biodegradability, and antibacterial ability when used as a component of composite material [26]. Furthermore, Hap is relatively stable under the change of pH medium or temperature.

This research proposes the nanocomposite based on SPIONs and hydroxyapatite composite as a magnetic drug carrier for an anticancer drug delivery platform. As the physicochemical properties of nanostructures strictly depend on the experimental conditions during synthesis, we presented the optimal parameters to prepare the material that can be used in magnetic hyperthermia. Such a technique can be used for cancer treatment for the local increase in the temperature of the target tissue from 42 to 46 °C within the application of SPION-based materials acting as heat mediators [27]. The main advantages of magnetic hyperthermia are the ability to increase the heating power of the nanoparticles, control the local temperature of the tumor, and damage local cells, and that such a technique can also be combined with the local drug delivery without affecting healthy tissues. Following this trend in this paper, the functional nanocomposite that can be used in magnetic hyperthermia is proposed.

Since the literature widely proposes several experimental conditions during syntheses, including different temperatures and different precipitating agents such as ammonia NaOH, or KOH [28–33], in this work, these parameters, as well as the final pH of the solution, were changed. Therefore, we have focused on optimizing the experimental conditions during synthesis and following coating with hydroxyapatite in different % wt. to obtain a spherical composite with a size below 20 nm and superparamagnetic properties. The nanocomposite containing up to 20% wt. of HAp coating SPIONs was proposed as the functional platform for the drug delivery system.

## 2. Experimental

### 2.1. Materials and Methods

Iron chloride tetrahydrate $FeCl_2 \cdot 4H_2O$, and iron chloride hexahydrate $FeCl_3 \cdot 6H_2O$ were obtained from Merck Company (Darmstadt, Germany). Ammonia solution (25%) and

citric acid were purchased from Guangdong Guanghua Sci-Tech Co. Calcium dinitrate $Ca(NO_3)_2 \cdot 4H_2O$, and ammonia monohydrogen phosphate $(NH_4)_2HPO_4$ were supplied from XilongScientific Co. All chemicals were analytical grade and used without purification. Acetone and sodium hydroxide NaOH were supplied from Merck (Darmstadt, Germany). Deionized water distilled within the Milli-Q filtering system was used in all of the experiments.

### 2.2. Synthesis of SPION

Superparamagnetic iron oxide nanoparticles (SPIONs) were prepared within the coprecipitation method from aqueous $Fe^{2+}/Fe^{3+}$ solutions. Briefly, $FeCl_2 \cdot 4H_2O$ (198.9 mg) and $FeCl_3 \cdot 6H_2O$ (540 mg) were dissolved in deionized water (DI) (10 mL) and stirred magnetically (600 rpm) for 10 min. Then, the precipitating agent, such as $NH_3$ or NaOH solution, was slowly dropped into the aqueous solution of $Fe^{2+}$ and $Fe^{3+}$ to reach high pH level in 30 min with vigorous magnetic stirring under the nitrogen atmosphere. The obtained solution was washed several times with DI water to remove the excess $NH_3$ and chloride. Then, SPIONs were collected by magnet and washed through neutral pH. Within the synthesis, the precipitating agent, such as $NH_3$ or NaOH, temperature of solution, such as 30 °C, 50 °C, and 80 °C, and pH such as 9.5, 10.09, and 11.1, were adjusted during the synthesis.

### 2.3. Synthesis of SPION/HAp

After preparation, SPIONs were modified with citrates (acid citric 0.1 M + NaOH to adjust pH 5–5.5). The mixture was stirred at 1000 rpm (30 min) and 600 rpm for the next 30 min at 60 °C. The solution was deposited, decanted, and washed with acetone to remove excess citric acid. SPIONs coated with citrates (SPION/CA) were collected on a magnet after washing with acetone while they were not dried. Then, HAp nanoparticles were synthesized by using $Ca(NO_3)_2 \cdot 4H_2O$ and $(NH_4)_2HPO_4$, where 200 mg of SPIONs (still wet with traces of acetone) were mixed gently with $Ca(NO_3)_2 \cdot 4H_2O$ solution and then for a few minutes. Then, $(NH_4)_2HPO_4$ was added dropwise along with $NH_3$ with the pH adjusted 10–10.5 during HAp precipitation, where phosphate-based solution and ammonia solution were placed in separate burettes. The chemical reaction was performed in 30 min and stirred at 800 rpm. Removing all the excess $NH_3$ to reach neutral pH by centrifugation. The SPIONs/HAp were obtained after drying in a vacuum medium. The dried samples were ground into powder by mortar. The HAp content was estimated based on the total mass of the product before and after the coating.

### 2.4. Method and Equipment

The morphology of SPION and SPION/HAp nanocomposites were examined by scanning electron microscope (SEM, Hitachi S-4800-Japan) and transmission electron microscope (TEM, Jem 1400 Flash, Jeol-Japan). The ImageJ software was used to analyze the size of nanoparticles based on the TEM images. The element composition of these samples was determined by SEM-EDX (SM-6510LV, Jeol-Japan, X-Act, Oxford Instrument) (Oxford, UK). The saturation magnetization of SPIONs was measured by a homemade vibrating sample magnetometer (homemade VSM) under the maximum applied field of 11 kOe at room temperature. Fourier transform infrared (FTIR) spectra of samples were recorded by Nicolet iS10, Thermo Scientific (New Castle, PA, USA). X-ray diffraction was obtained by D8 ADVANCE-Bruker, CuKα radiation (λ = 1.54056 Å) with step angle of 0.030°, the scanning rate of 0.04285° $s^{-1}$, and 2θ degrees in the range of 10–70°. The magnetic hyperthermia measurements were performed with an equipment SI-10KWHF supplied from the Superior Induction Company (Pasadena, CA, USA), where the temperature was determined with FLIR A300.

## 3. Results

### 3.1. Characterization of SPION

#### 3.1.1. Effect of Experimental Conditions during Synthesis on Magnetic Properties

As the critical parameter that determines the potential of composite use for magnetic hyperthermia is the saturation magnetization and superparamagnetism, these features were initially determined for bare SPIONs within the vibrating magnetometer (VSM). The measurements were performed in 300 K. SPIONs are precipitated within the alkali medium, the two different compounds being a source of hydroxyl groups—an aqueous solution of $NH_3$ and NaOH. Since both compounds have alkalic properties, where the -OH groups are responsible for the precipitation of the SPIONs from the solution, no change in the magnetic properties was expected. Figure 1a shows the magnetization curves for SPIONs precipitated with these agents in the function of the magnetic field, where the synthesis was performed at 30 °C. As can be seen, the magnetization saturation ($M_s$) was higher when ammonia was used. The magnetization of SPIONs synthesized by $NH_3$ reached approximately 56.2 emu $g^{-1}$, whereas using NaOH reached only 45.2 emu $g^{-1}$. Inset shows the similar coercivity for both conditions. Inset shows similar coercivity for both conditions. Literature also indicates the co-precipitating agent effect on the magnetic properties [34]. Based on these results, the $NH_3$ was proposed as a co-precipitating agent for the final pH effect during precipitation studies. After the ammonia solution addition, the pH slightly decreased during synthesis for the incorporation of the hydroxyl groups in the SPIONs, so the pH was maintained for the whole synthesis to remain constant. The endpoint was proposed as follows: pH 9.5; 10.09; and 11.1. With the highest pH 11.1, the magnetic saturation point reached approximately 65.0 emu $g^{-1}$, significantly enhanced in comparison with the lower pH 9.5 and 10.09 that were 60.71 emu $g^{-1}$ and 60.47 emu $g^{-1}$, respectively. Next, the synthesis was performed in 30 °C, 50 °C, and 80 °C. Figure 1c presents the slight improvement of the $M_s$ with an increase in the temperature. At 80 °C, magnetization saturation reached approximately 66.3 emu $g^{-1}$, while it reached only 61.68 emu $g^{-1}$ recorded in 30 °C. The inset presents a very narrow coercivity for the samples, where the narrowest is observed for the nanoparticles synthesized with the highest temperature. Figure 1d shows the graph for the conditions that were proposed for the following coating with hydroxyapatite: ammonia solution as a precipitating agent, pH 11.1, and 80 °C. The inset shows that no energy dissipated during repeated magnetization reversing, confirming the prepared material's superparamagnetic character. Therefore, based on the magnetic properties, the highest pH and temperature were chosen for the following HAp coating. $M_s$ values are in good agreement with the data presented in the literature [35,36].

#### 3.1.2. Crystallinity of SPIONs

Next, the crystallinity of the SPIONs was determined within the X-ray diffraction (XRD) method in the 2θ angle range from 10 to 70°. XRD patterns were recorded for the samples prepared for the 30 °C, 50 °C, and 80 °C in pH 11.1. All the samples exhibited the typical crystalline of $Fe_3O_4$ (JCPDS card 02-088-0315). As can be seen in Figure 2, in all samples the characteristic peaks at 30.1°, 35.4°, 43.1°, 53.5°, 57.1°, and 63.2° can be distinguished. They can be ascribed to the (220), (311), (400), (422), (511), and (440) planes. Recorded patterns are in good agreement with the literature, confirming the presence of bare $Fe_3O_4$ in all samples [37,38].

(**a**)

(**b**)

(**c**)

(**d**)

**Figure 1.** Magnetization changes in function of an amplitude of the magnetic field for (**a**) different co-precipitation agents: $NH_3$ and NaOH, (**b**) different pH: 9.5; 10.09; 11.1, (**c**) different temperature during synthesis 30 °C, 50 °C, and 80 °C, and (**d**) optimized conditions.

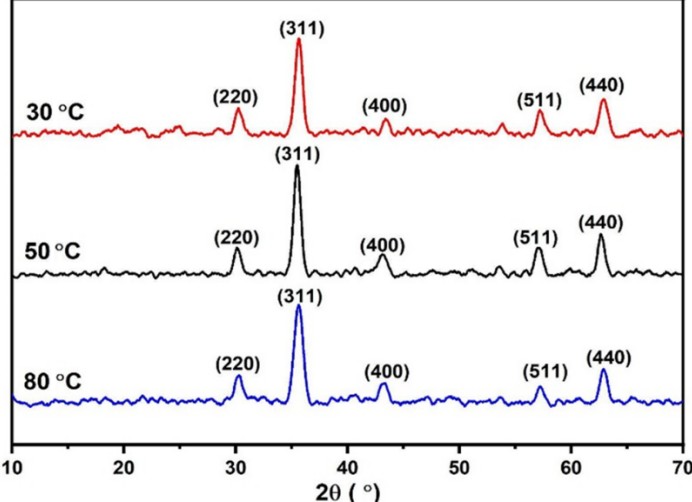

**Figure 2.** XRD patterns of SPIONs recorded at 30 °C, 50 °C, and 80 °C.

### 3.1.3. Morphology Studies

The morphology and structure of synthesized nanoparticles were evaluated by scanning electron microscopy (SEM). The measurements were performed for the samples synthesized under the ammonia use at different pH, while no significant change in the morphology was expected. Within the whole samples, the shape of nanoparticles is relatively uniform in that the average size of SPIONs prepared is approximately 15 nm in most of the points of pH levels measured, see Figure 3. The agglomeration is caused by the drying of SPIONs on the carbon tape. Within all samples, the uniform distribution of particles can be observed.

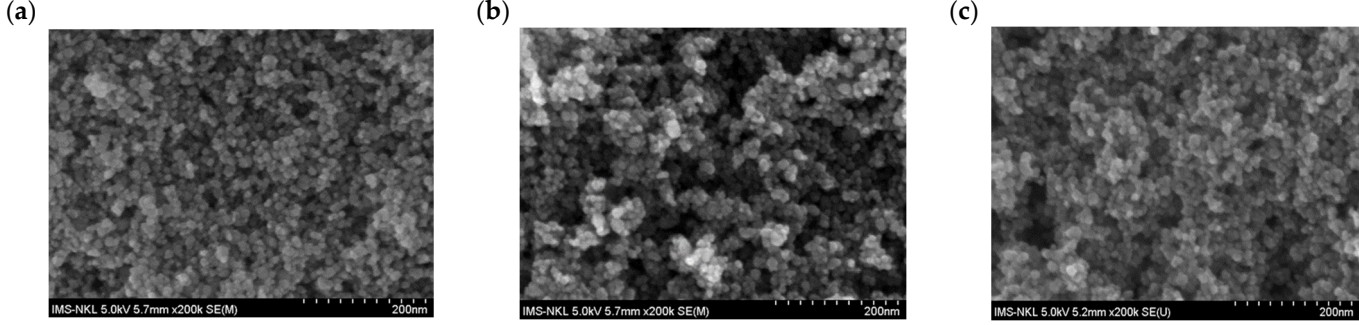

**Figure 3.** SEM images of SPIONs recorded at pH (**a**) 9.5, (**b**) 10.09, and (**c**) 11.1.

As the optimal temperature proposed based on the magnetic studies was 80 °C, the morphology was investigated for the sample prepared particularly at this temperature at pH 11.1. Complementary to the SEM images, the transmission electron microscopy (TEM) was used to determine the shape and size of the nanoparticles. Additionally, the energy dispersive X-ray diffraction was used. Figure 4a shows a TEM image of bare SPIONs and Figure 4b illustrates the size distribution of the SPIONs (based on the TEM image in a selected region). The TEM image reveals a mean diameter of 12 nm in size and the round shape of SPIONs. The obtained magnetic $Fe_3O_4$ nanoparticles are relatively uniform and almost spherical.

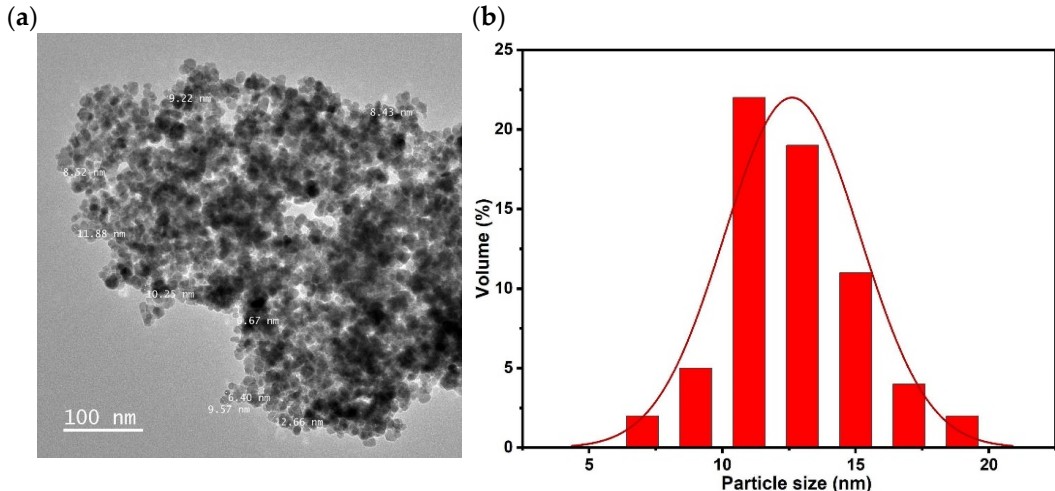

**Figure 4.** (**a**) TEM image and (**b**) size distribution of SPIONs synthesized at 80 °C in pH 11.

### 3.2. Modification of SPION by Citric Acid

Next to the synthesis and characterization of SPIONs, they were modified with citrates to improve their stabilization. The following Figure 5a presents the results from the DLS, where the average hydrodynamic size of SPIONs is 44.7 nm and the 82.3 nm for SPIONs stabilized with citrates. The zeta potential was estimated after the stabilization of

approximately -37.2 mV. The composition was investigated with the FT-IR spectroscopy for bare SPIONs and after modification with citrates (SPION/CA), see Figure 5b. The peak recorded at 575 cm$^{-1}$ represents the stretching vibration Fe-O in the crystallographic lattice of SPIONs. The following sharp peak at approximately 1625 cm$^{-1}$ and the broad band at 3400 cm$^{-1}$ can also be ascribed to the vibration in the Fe-O, including hydroxyl groups [34,39] such as the angular vibration of O–H and O–H stretching [40]. After coating, the peak of Fe-O was slightly shifted from 575 cm$^{-1}$ to 616 cm$^{-1}$, which stands for the interaction between $Fe_3O_4$ and citric acid. The minor peak at approximately 1395 cm$^{-1}$ is characteristic for deprotonated carboxylic groups (COO$^-$) for SPION/CA. The peaks at around 3600 cm$^{-1}$ and 3150 cm$^{-1}$ represent O-H intermolecular and intramolecular hydrogen bonds [41]. Citrates were used as a calcium phosphate inhibitor that induces Hap formation [42]. Due to the citrate's application, SPIONs can be coated uniformly within the whole surface.

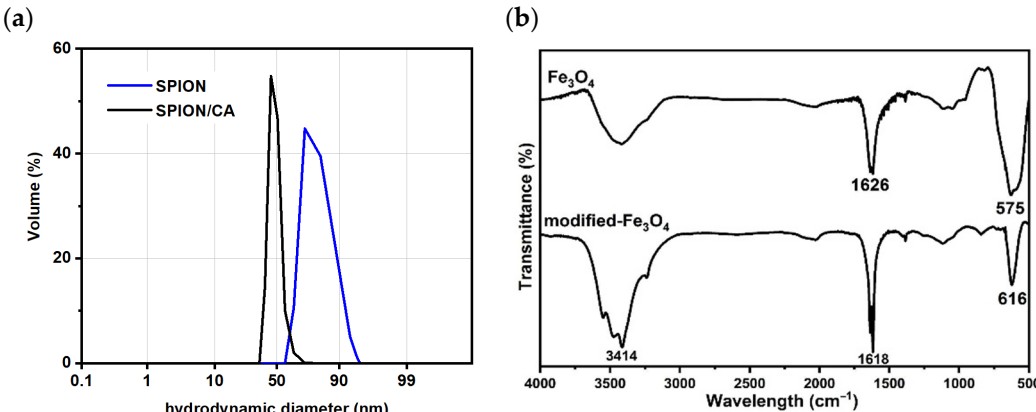

**Figure 5.** (**a**) DLS analysis (**b**) FT-IR analysis of SPIONs before and after modification with citrates.

### 3.3. Characterization of SPION/HAp Nanocomposite

### 3.3.1. Magnetic Behavior of Nanocomposite

Next, the SPION/CA was coated with hydroxyapatite for its following application in magnetic hyperthermia. Therefore, the magnetic properties of the SPION/HAp nanocomposite containing different amounts of HAp were investigated. Figure 6 shows the magnetization curves for the composite, where SPIONs consisted of 90% to 5% of the total weight of the composite. The increase in the magnetic core amount in the composite also increases the $M_s$. However, the role of HAp was to act as a porous platform for drug delivery, so its different amount in nanocomposite was tested. With a 10% mass of SPION, the $M_s$ is approximately 4.69 emu g$^{-1}$ and even less for 5% of SPIONs. It can be seen that even a 10% HAp content in the composite leads to a decrease in $M_s$, so for the magnetic hyperthermia, only the samples above 80% of SPIONs wt% content were chosen. The results obtained for the proposed composites resemble similarities within the values of the magnetic saturation of composites described in the literature [43], what is a crucial factor for the biomedical studies. Some studies refer the broader hysteresis loop for SPION/HAp based composite [44], so the data obtained in this work are promising for the following magnetic hyperthermia studies.

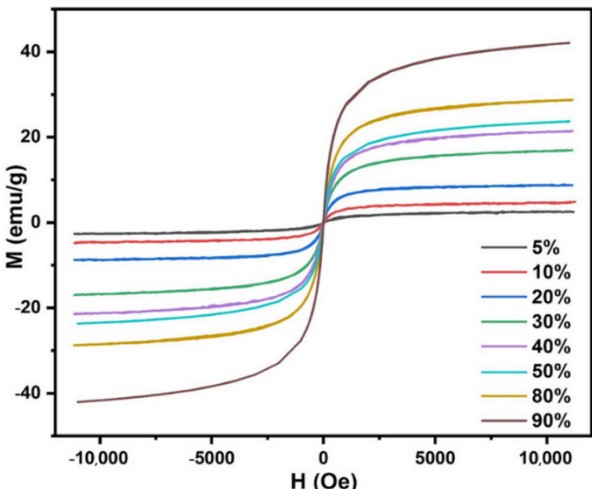

**Figure 6.** Magnetization curves of nanocomposites with different ratios SPION/HAp at 80 °C with pH 11.1.

### 3.3.2. Crystallinity of Nanocomposite

XRD patterns of nanocomposite followed by the percentage of mass of SPION is shown in Figure 7. All peaks belonging to the presence of hydroxyapatite and $Fe_3O_4$ have appeared. The diffraction peaks represented for $Fe_3O_4$ become more clear and for Hap, which is decreased with the increase in $Fe_3O_4$ concentration. Peaks representing the functional group of $Fe_3O_4$ and HAp nanoparticles are varied with the concentration of particles. The most dominant peak for calcium phosphate/hydroxyapatite-coated nanoparticles was obtained at $2\theta = 32.4°$, corresponding to the (211) crystal plane, as well as the peak at $2\theta = 26.8°$ and $53.4°$ corresponding to the (002) and (422) crystal planes marked with the * symbol. These patterns are in good agreement with the literature [45,46]. From the values obtained by the XRD analysis, the lattice parameter and crystallite size of SPION/Hap were calculated to be 8.4 Å for magnetite using the peak at 35.4° belonging to the (311) crystal plane [47]. Obtained results for the SPION/Hap structure are in good agreement with the results presented in the literature for, e.g., HAp/SPION structure [48,49].

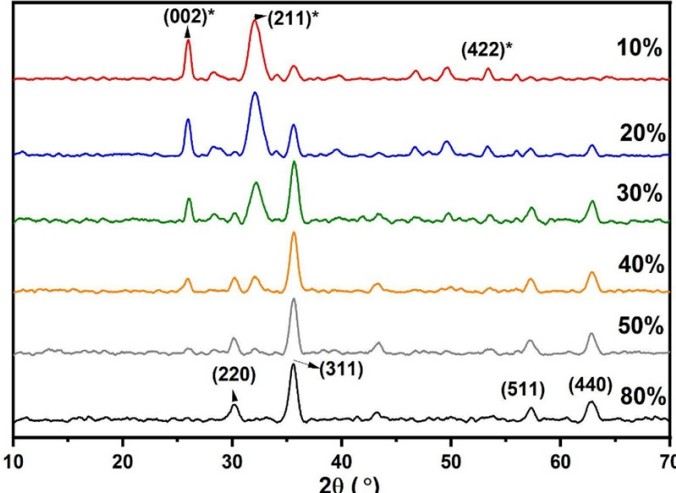

**Figure 7.** XRD pattern of nanocomposites with different ratios of SPION/HAp at 80 °C with pH 11. Symbol * corresponds to the HAp phase.

### 3.3.3. Nanocomposite Morphology Studies

Analogically to the morphology studies of bare SPIONs, the composite was also investigated with SEM. Figure 8 shows the influence of HAp content on the shape and size

of structures formed by the composite. With the increase in the HAp concentration having a rod-like shape [50,51], it can be seen that Fe$_3$O$_4$ nanoparticles might be easily covered by more rod-HAp. As seen in Figure 8a–c, corresponding to the composite containing 90%, 80%, and 70% of HAp by mass, respectively, the morphologies are commonly rod-like structures. Such a morphology is caused by a too-thick coating layer of HAp causing agglomeration and non-regular growth onto SPION. The 60% and 50% HAp were coated over the sphere of Fe$_3$O$_4$ nanoparticles that were presented in Figure 8d,e. Following Figure 8f,g reveals a granular structure, where the graphs present a 30% and 20% HAp coating and bare SPION. As can be seen, the coating of approximately 20–30% has the most uniform shape and size of particular structures.

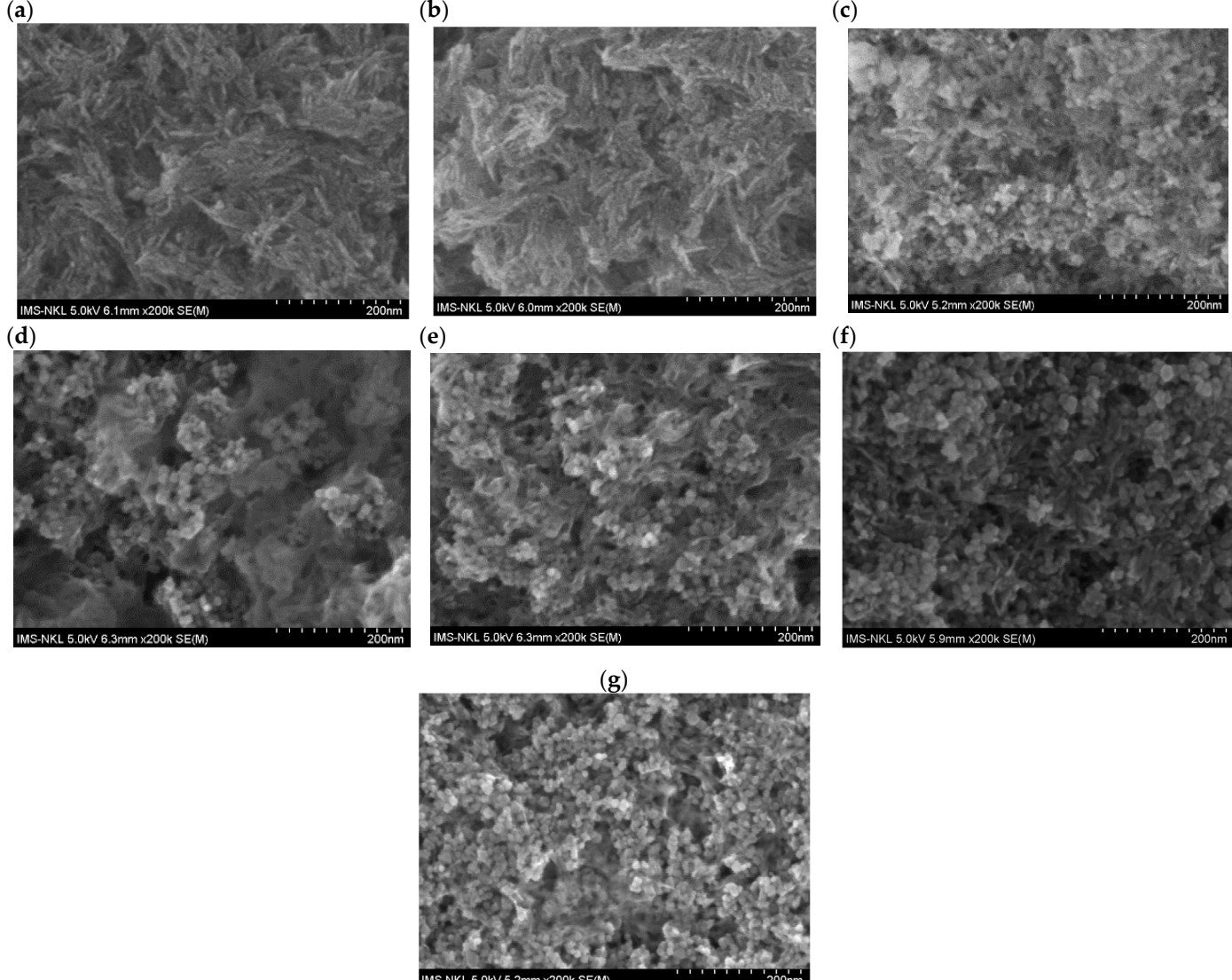

**Figure 8.** SEM images of SPION/HAp prepared at 80 °C with pH 11, where HAp consists (**a**) 90%, (**b**) 80%, (**c**) 50%, (**d**) 40%, (**e**) 30%, (**f**) 20%, and (**g**) 10% of wt% over SPIONs in nanocomposite.

Next, the chosen samples containing 80% and 90% of SPIONs were studied with transmission electron microscopy (TEM), see Figure 9. It is clearly seen that the spherical SPION core is coated with Hap, forming a shell similarly to the results described in the literature [52].

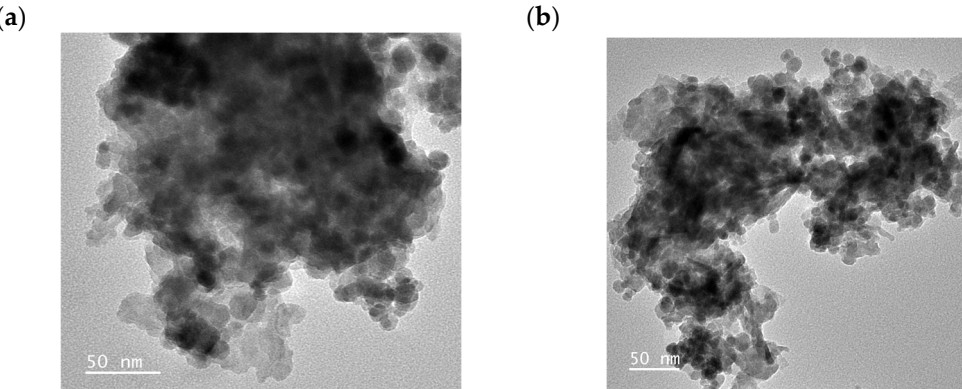

**Figure 9.** TEM images of a SPION/HAp nanocomposite sample containing (**a**) 80% and (**b**) 90% of SPIONs.

Complementary to the SEM analysis, the EDS measurements were performed to confirm the presence of particular elements of HAp in the composite. It can be seen from EDS analysis that $Fe_3O_4$ nanoparticles give apparent Fe- and O-related peaks where no other contamination-related peaks can be seen. The EDS results of SPION/HAp nanocomposite are presented in Figure 10b,c, where the particular peaks for Ca, Fe, and O can be seen, where the content of SPIONs in the composite was 50% and 80%, respectively.

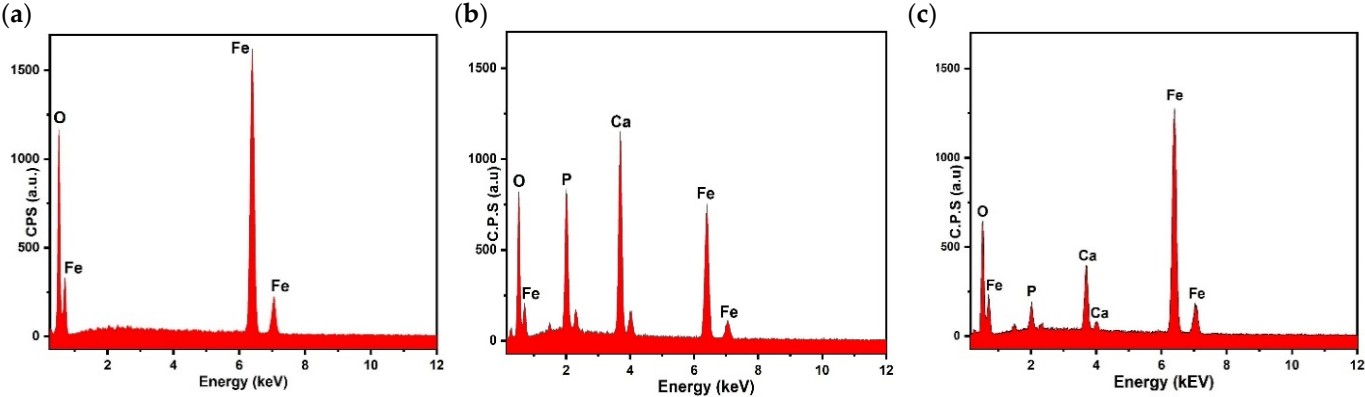

**Figure 10.** EDS spectra for the (**a**) bare SPION, (**b**) 50% SPION/HAp, and (**c**) 80% SPION/HAp.

The mass and atomic % of the particular elements in the $Fe_3O_4$ and $Fe_3O_4$/HAp are presented in Table 1. As the concentration of HAp rises, the signals for Ca and P in the composite increase.

**Table 1.** EDS result of SPIONs and SPION/HAp composite.

| Nanoparticle Type | Fe (%) | | O (%) | | Ca (%) | | P (%) | |
|---|---|---|---|---|---|---|---|---|
| | %m | %a | %m | %a | %m | %a | %m | %a |
| Bare SPION | 39.67 | 69.66 | 60.33 | 30.34 | - | - | - | - |
| 50% SPION/ HAp | 29.13 | 13.00 | 43.37 | 63.31 | 16.53 | 10.23 | 10.46 | 3.42 |
| 80% SPION/HAp | 61.57 | 34.76 | 29.06 | 57.27 | 6.74 | 5.30 | 2.63 | 2.67 |

### 3.3.4. DLS and FT-IR ANALYSIS

Figure 11a shows the DLS analysis of SPION/HAp samples tested within the magnetic hyperthermia and Figure 11b the FT-IR result of nanocomposite with an ascending order of SPIONs concentration. The hydrodynamic size is similar to the results described in the literature for the magnetic HAp composites [53,54].

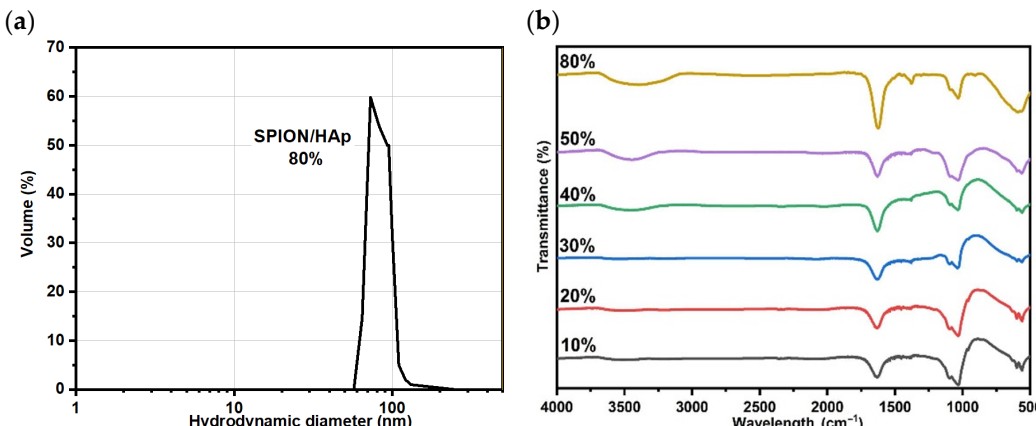

**Figure 11.** (**a**) DLS analysis of SPION/Hap and (**b**) FT-IR spectra of nanocomposite with different ratio SPION/HAp at 80 °C with pH 11.

According to FT-IR, as expected, the peaks were observed at 563 cm$^{-3}$ and 601 cm$^{-3}$, indicating the formation of the metal–oxygen Fe-O bonds in the crystalline lattice of $Fe_3O_4$, just like in the bare SPIONs. The vibrations at 1038 cm$^{-1}$ and 1637 cm$^{-1}$ represent the asymmetric stretching mode of vibrations of the functional group $PO_4^{3-}$ and O-H stretching of HAp nanoparticles [55–58]. It can be seen that the more the SPIONs concentration ascends in the nanocomposite, the more that the vibrations of phosphate groups represented for HAp decrease.

3.3.5. Magnetic Hyperthermia Analysis

To determine the potential of the proposed nanocomposite for its use in magnetic hyperthermia (MH), the suspension was treated with an alternating magnetic field, and the temperature increase in the function of time was recorded. The sample that was proposed to be used in an MH was nanocomposite SPION/HAp containing 80% of SPIONs wt.%. The targeted temperature generated within the alternating magnetic field (AMF) was 42–46 °C, which can be maintained within several minutes. Initially, the samples were suspended in the Milli-Q water, where the concentration of SPION/HAp was approximately 10 mg L$^{-1}$, the frequency was 340 kHz, and the magnitude of the field was 250 G, see Figure 12a. As can be seen, samples easily heat above 50 °C. Following Figure 12b shows the temperature change of samples in the cultivation medium for cells growth. The measurements show that the suspension easily reaches the expected temperature. After these initial tests in AMF with the point-temperature probe, the measurements were followed in different AMF conditions using the thermovision camera as a temperature detector, where the sample was placed on the Petri dish, see Figure 12c. In all cases, the SPION/HAp (80% SPIONs) were used.

The measurements were performed on a Petri dish (diameter of approximately 50 mm) with 20 mg mL$^{-1}$, where the heat distribution while heating the SPIONs was recorded. As can be seen in Figure 13a, the suspension had room temperature when the AFM was generated for approximately 5 s. The yellow ring presents the coil. Figure 13b shows the heat generation under the heating, where the colloidal suspension warmed up by approximately 4 °C after 20 s of heating and reached approximately 43 °C after 90 s of AMF generation with the 540 kHz and 21 A, but still, the temperature increased to 43 °C, see Figure 13c. Therefore, the amplitude was reduced to 19 A, 17 A, and 15 A, where within 90 s it reached approximately 41 °C (see, Figure 13d–f, respectively), but still, raised up at 47 °C. Therefore, the AMF was modulated to reach 43 °C, and then, it was maintained with a lower amplitude of approximately 6 A. At this point, the different parameters to maintain 42–46 °C were estimated. As too large values of AMF may generate the eddy current, the high amplitude should be reduced to keep the parameters as low as possible

with the therapeutic temperature. Measurements were performed under the conditions below the Atkinson–Brezovich limit.

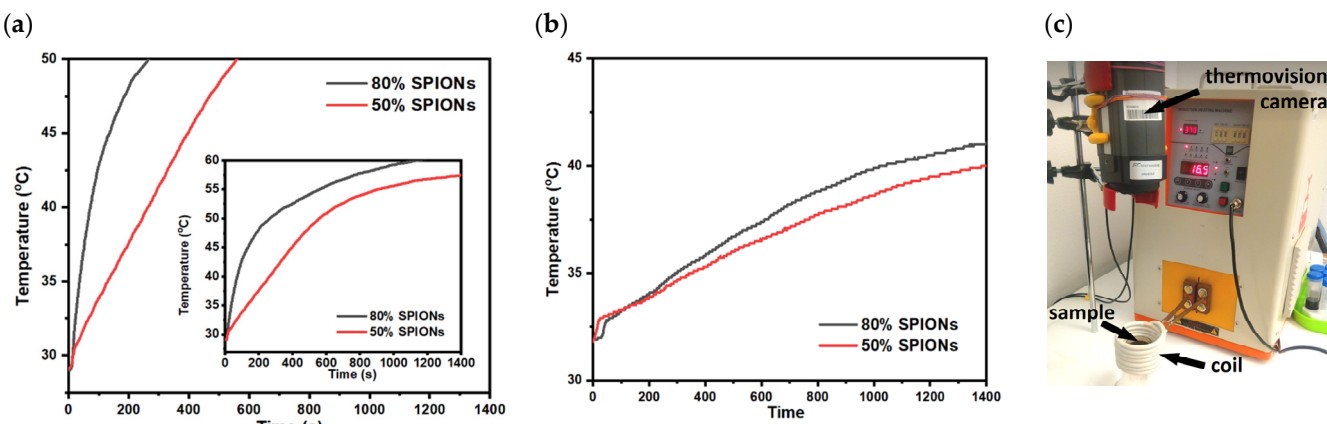

**Figure 12.** Temperature of nanocomposites magnetization with different concentrations of SPIONs and solvent, respectively, (**a**) 5 mg mL$^{-1}$ in DI water, (**b**) 10 mg mL$^{-1}$ in solution in cultivation medium, and (**c**) magnetic hyperthermia setup.

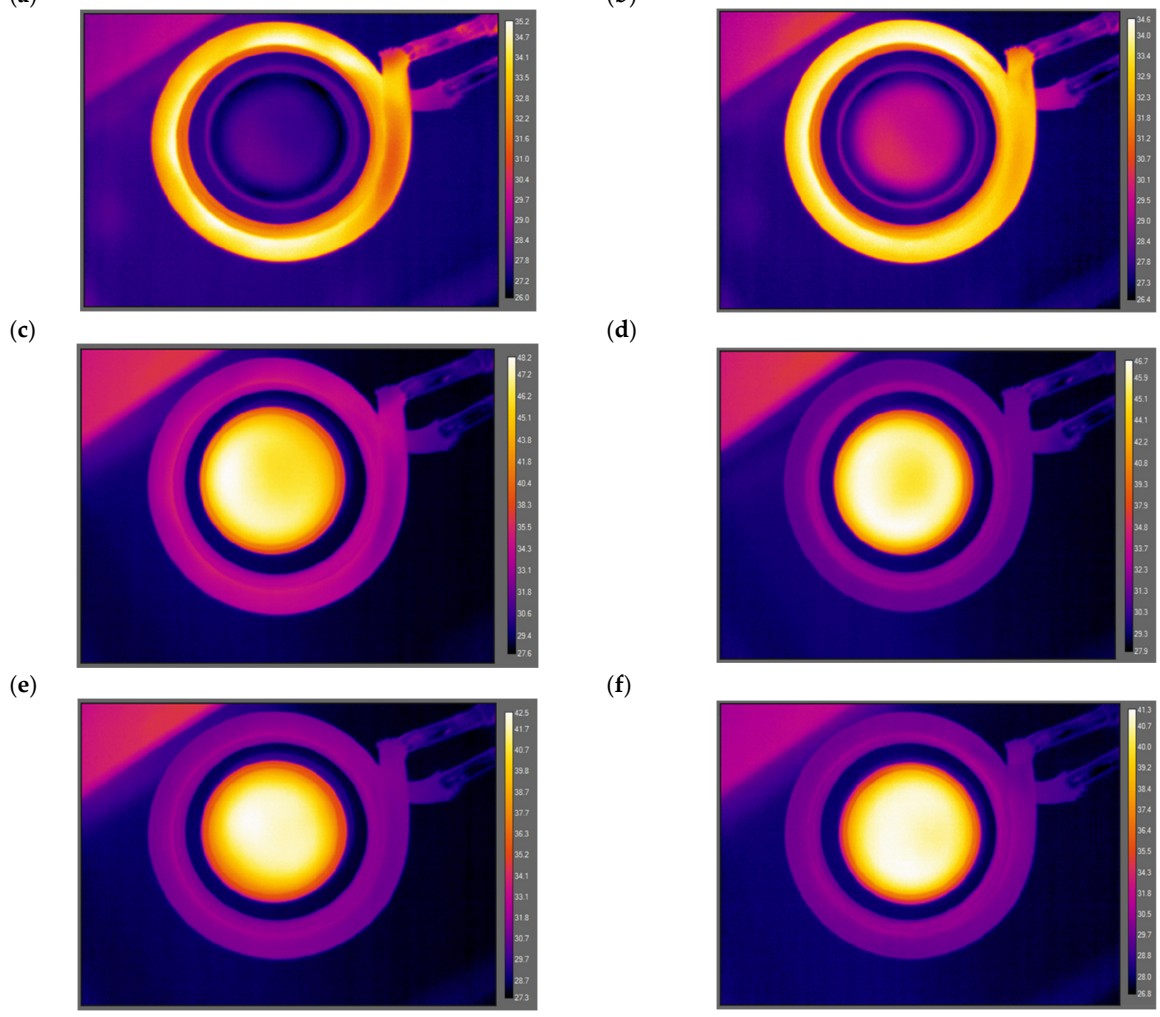

**Figure 13.** SPION/HAP under the AMF 540 kHz, 21 G, after (**a**) 5 s, (**b**) 30 s, (**c**) 90 s of AMF with retain 6 G, (**d**) 19 G, (**e**) 17 G, and (**f**) 15 G.

The change of the temperature in the function of time for the SPION/HAp under the AMF with the frequency of 540 kHz and the amplitude of approximately 15 G, 17 G, 19 G, and 21 G is presented in Figure 14. The measurements were performed for 10 min, where the amplitude of the AMF was varied. Depending on the concentration of the nanocomposite, the heating temperature slightly varied, while in both cases of 15 mg mL$^{-1}$ and 20 mg mL$^{-1}$, the therapeutic temperature was reached. The results are comparable with the nanocomposite based on SPIONs and HAp in a different ratio [49].

(**a**)        (**b**)

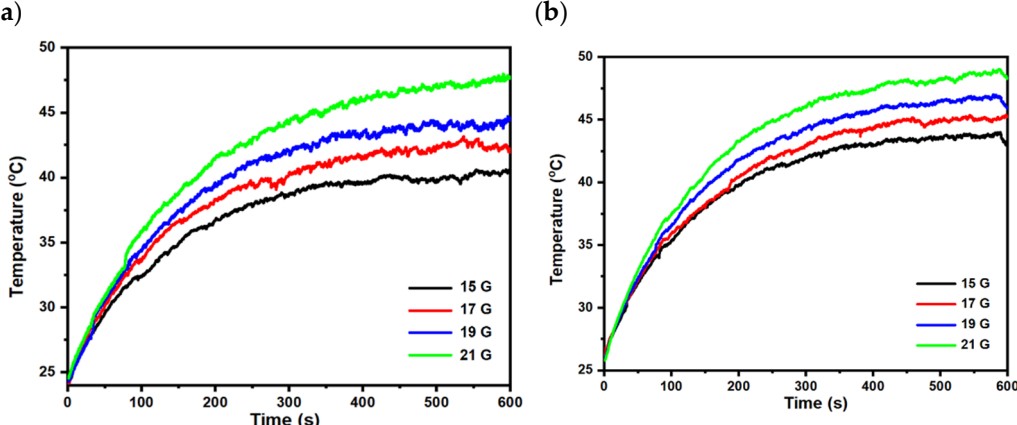

**Figure 14.** Temperature change in time for 15 mg mL$^{-1}$ (**a**) and 20 mg mL$^{-1}$ (**b**) for SPION/HAp under the AMF, where f = 540 kHz.

Then, the specific absorption rate (*SAR*) was estimated based on the following equation:

$$SAR = \frac{d \cdot C_e}{N_p} \left( \frac{dT}{dt} \right)_{max}$$

where the *d* is the dispersant density (kg·m$^{-3}$); $C_e$ is a dispersant specific heat (kcal·kg$^{-1}$ $^\circ$C$^{-1}$); N$_p$ is the nanoparticles density (kg·m$^{-3}$); *T* is the temperature ($^\circ$C), and t is the time (s). The values for the *SAR* for the proposed conditions at 540 kHz and 15–20 G are approximately ~33–48 W·g$^{-1}$ for the nanocomposite containing 20% of HAp by weight. The results are comparable with the literature where for example nanocomposite containing below 10% of HAp in the structure generated 85 W·g$^{-1}$ [59].

Additionally, the measurements where the amplitude of the magnetic fields was applied by pulses were performed. As can be seen in the following Figure 15, the increase in the temperature in the function of time was recorded. In this part of the experiment, the pulses of AMF having different values were applied in different timings. It can be seen in Figure 15a that depending on the pulse duration, the temperature of the suspension can be regulated to maintain the therapeutic temperature. It is seen that within the increase of the pulse duration the temperature also increases. Within the smaller changes of the amplitude in shorter periods than for Figure 15b, the therapeutic temperature can be easily obtained, see Figure 15b. The application of the AMF having different amplitudes within pulses can be helpful within the biological studies, where the reduction of the amplitude with a maintained heat efficiency is desired.

(**a**)                                        (**b**)

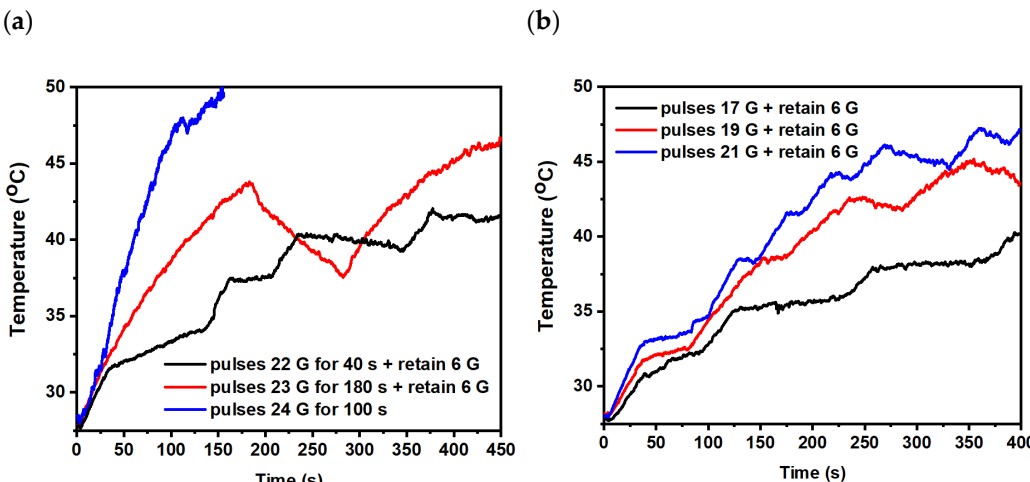

**Figure 15.** Temperature change in time for 20 mg mL$^{-1}$ where the AMF was applied with different amplitude by pulses (**a**) 22–24 G with retain 6 G and (**b**) 17–21 G with retain 6 G, where f = 540 kHz.

## 4. Discussion

This work aimed to prepare the nanostructural composite based on the SPION coated with hydroxyapatite through the co-precipitation method as a platform for drug delivery and magnetic hyperthermia. The experimental conditions were changed within the synthesis to determine the optimal parameters to obtain spherical structures sized below 20 nm and superparamagnetic properties. So far, the influence of the precipitating agent, pH, and temperature on the physicochemical properties of the magnetic core has been explored. It was found that the ammonia solution used instead of NaOH improves the saturation magnetization of the particles. The optimal endpoint of the precipitation was determined as approximately pH ~11, and the temperature of approximately 80 °C. At these conditions, the particles have superparamagnetic properties with an $M_s$ of approximately 69 emu g$^{-1}$ with a round shape and a size below 15 nm. Then, after the optimization of the experimental conditions, SPIONs were coated with citrates, improving the colloidal suspension's stability. Furthermore, the suspension was coated with hydroxyapatite, where the amount of HAp onto the SPIONs surface was modified in the range from 10% to 90% wt.%. With the increase in the HAp content on the SPIONs, the magnetization of the nanocomposite decreased, and the HAp formed non-regular structures of even agglomerates. Based on the characterization, the optimal composition to obtain a nanocomposite with a high magnetization, a spherical shape, and a size below 20 nm was estimated to be approximately 20% HAp wt.% Then, the optimized nanocomposite was tested with magnetic hyperthermia (MH) to check if the suspension can generate heat under the alternating magnetic field, making it possible to be used in the future for biomedical applications. A therapeutic temperature of approximately 42–46 °C in the aqueous media can be reached under the application of AMF with a constant value of amplitude or within pulses, making it possible to regulate the therapeutic temperature within the pulses. A pulsed application of the AMF can be safer than the application of a constant amplitude of AMF. Nevertheless, besides the MH studies, it is needed to perform the studies in the function of the ionic strength and different media and in the presence of different biomacromolecules such as proteins as well as media having different viscosities. Following step of the studies will be MH studies in different ionic strengths, viscosities, and implementations of the anticancer drug and its release under the AMF condition and in the function of time.

## 5. Conclusions

In this study, SPIONs were modified and coated with HAp nanoparticles along with the SPIONs/HAp nanocomposite being synthesized at different ratios. The nanocomposite samples were characterized by various techniques that include SEM, TEM, VSM, FT-

IR, EDX, XRD, and MH analysis. The magnetic properties of SPIONs have changed by the effect of temperature, pH medium, and precipitating agents, while the highest $M_s$ values were obtained for SPIONs prepared at 80 °C, pH~11 with $NH_3$, and precipitating agent. As the amount of the HAp in the nanocomposite influences the morphology and magnetization, the optimal amount of HAp was proposed as 20% of wt.%. A higher content of HAp drastically affected the morphology, leading to the agglomerate's formation in the sample. Therefore, the samples proposed for the further application cannot exceed the 20% by mass of the proposed composite. Magnetic hyperthermia studies proved that the colloidal suspension of SPION/HAp can generate heat under the alternating magnetic field reaching a therapeutic temperature of approximately 42–46 °C, where the temperature can be generated within the pulses having different amplitudes of AMF (with different pulses lasting). The proposed nanocomposite can be used for magnetic hyperthermia, where HAp can be also used as a platform for drug loading and further release under the AMF. The limitations are the safe biological conditions determined within the Brezovich limit and the size of the composite, where the increase in the size affects the heat generation efficiency. Further studies will attempt to stabilize the SPION/HAp suspension with the organic compounds and conjugating with anticancer drugs for the drug release under the AMF for the multiplied therapeutic effect of local chemotherapy and magnetic hyperthermia. Prior to the bioconjugation, the stabilization needs to be optimized to minimize the agglomerates formation, especially under the AMF.

**Author Contributions:** Conceptualization, D.T.M.T. and M.O.; Methodology, D.T.M.T., M.O.; Formal Analysis, N.T.P., D.T.H., H.N.G., N.T.T., P.T.N., N.T.D., M.G., M.O.; Investigation, all authors; Writing—Original Draft Preparation, H.N.G., N.T.P., M.O.; Visualization, M.O., Writing—Review and Editing—all co-authors; Supervision, M.O., D.T.M.T.; Funding Acquisition, D.T.M.T. All authors have read and agreed to the published version of the manuscript.

**Funding:** This research was funded by the Vietnam Academy of Science and Technology (VAST) with grant no. CT0000.09/21-23.

**Acknowledgments:** M.O. would like to thank Hubert Grzywacz from IPPT PAN for technical consultations. M.O. would like to thank to Piotr Jenczyk from IPPT PAN for SEM analysis support.

**Conflicts of Interest:** The authors declare no conflict of interest.

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
