# Peer review of "Influence of Experimental Conditions during Synthesis on the Physicochemical Properties of the SPION/Hydroxyapatite Nanocomposite for Magnetic Hyperthermia Application"

_magnetochemistry, doi:10.3390/magnetochemistry8080090_

Round 1

Reviewer 1 Report

The introduction of the manuscript shows that the authors are familiar with kinds of methods of SPIONs modification for improving biocompatibility and functionality. On basis of the background, they proposed the nanocomposite based on SPIONs and hydroxyapatite composite as a magnetic drug carrier. This manuscript presented the optimal parameters to prepare the material that can be used in magnetic hyperthermia, and might be used as a drug carrier though not proved in this research. The material preparation parameters proposed in this research provide a basis for future application, and also provide a reference for the preparation of similar materials. I suggest to accept after modification.

Nevertheless, the authors need to answer or modify the following questions

1 In Line 52 “…to induce the cancer cell and preserve the normal cell”, this sentence is inaccurate and not academic in cell biology. Does the authors mean "the local thermogenesis system can effectively damage tumor cells without harm to normal cells"?

2 Line 62, why "early cancer treatment", not benefit to middle and late stage cancer?

3 Figure 4. (b) shows size distribution of SPIONs. The conventional method to analyze size distribution is to use particle analyzer such as Malvern. Since the calculation was based on electron microscopy, how to select the particles to be measured and the number of particles measured need stated, or Malvern detection be supplemented.

4 As the different ratios given in Figure 6, how did the authors calculate or control the percentage mass of SPION? In other words, the ratios referred to proportion of components in the product SPION/HAp or the proportion of initial reactants, SPION and HAp?

5 In the part of magnetic thermal experiment, there is too little introduction to the experiment equipment, we hope to add a photo of the device, or at least a diagram. Also, it is necessary to explain how the temperature of the medium measured and how accurate was the thermometer.

6 The lack of cell mortality or viability testing in magnetic thermal effect and lack of efficacy in carrying drugs are somewhat unsatisfied. However, information should be given on the kind of the cell medium, the volume of the medium, and the diameter of the Petri dish used.

Author Response

We would like to thank you for your valuable comments and suggestions. We tried to include them all in our work.

Within the synthesis of the proposed composite, we have found out that depending on the synthesis conditions proposed in the literature, the results correlating the physicochemical properties of our composite with the experimental conditions differed from the literature. Therefore, we have decided to focus on the basic parameters described in the paper to show the effect of experimental conditions during synthesis on the properties of the composite.

Addressing the comments, we have corrected as manuscript as suggested.

1 In Line 52 “…to induce the cancer cell and preserve the normal cell”, this sentence is inaccurate and not academic in cell biology. Does the authors mean "the local thermogenesis system can effectively damage tumor cells without harm to normal cells"?

We meant temperature change induced by the thermogenesis of a high frequency alternating magnetic field. Thank you for this suggestion. We have modified the text.

2 Line 62, why "early cancer treatment", not benefit to middle and late stage cancer?

We meant that early addressed small-scale cancer towards the local temperature change. Thank you for this comment. We have modified the text. Recently, several sources proposed the application of MH even in the late stage, so we have corrected it.

3 Figure 4. (b) shows size distribution of SPIONs. The conventional method to analyze size distribution is to use particle analyzer such as Malvern. Since the calculation was based on electron microscopy, how to select the particles to be measured and the number of particles measured need stated, or Malvern detection be supplemented.

We have analyzed the size distribution based on the TEM using ImageJ software. It is included in the text now. Complementary, the hydrodynamic size was determined within the DLS Malvern equipment. The data are added to the manuscript.

4 As the different ratios given in Figure 6, how did the authors calculate or control the percentage mass of SPION? In other words, the ratios referred to proportion of components in the product SPION/HAp or the proportion of initial reactants, SPION and HAp?

The control was measured based on the mass of the final product. Initially, the mass of the samples after washing and drying was measured for particles without the HAp coat. Then, the mass of the HAp without modification of SPIONs’ surface was also weighed. After that, the final mass of the product SPION coated with HAp was weighed. Therefore, the amount of HAp over the total weight of the composite was measured. We included that in the draft.

5 In the part of magnetic thermal experiment, there is too little introduction to the experiment equipment, we hope to add a photo of the device, or at least a diagram. Also, it is necessary to explain how the temperature of the medium measured and how accurate was the thermometer.

Indeed, we missed that section in the manuscript. Thank you for finding the lack of the method description. We have added the information about the equipment as well as the image of the equipment within the magnetic hyperthermia section.

6 The lack of cell mortality or viability testing in magnetic thermal effect and lack of efficacy in carrying drugs are somewhat unsatisfied. However, information should be given on the kind of the cell medium, the volume of the medium, and the diameter of the Petri dish used.

In this part of the work, only the heating effect was investigated in different media having different viscosity than water. The measurements included in the MH studies included the cultivation medium, not the cells. Thank you for pointing this out in this paper; we have corrected the information about the medium. The diameter of the Petri dish was 50 mm - it is also added to the manuscript. We also included the SAR values.

Thank you once again for your valuable comments. Thank them we could inprove our work.

Reviewer 2 Report

The authors presented the paper "Influence on experimental conditions on the physicochemical properties of the SPIONs and hydroxyapatite for magnetic hyperthermia application"

1) The title of the paper is broad. I recommend clarifying the title of the article.

2) Much more 2-3 year review papers have to be used for Introduction section to show the perspectives of the area.

3) The novelty of the spions synthesis has to be mentioned. I have seen many papers about the synthesis using NH3 or NaOH (the base differences papers), and pH values, which shows differences in sizes, magnetic properties, etc. Is the synthesis section 2.2 and its properties (section 3.1) are new?

4) For the biological applications the sizes of the nanocomposites must be between 15-200 nm. However, I see SEM and TEM figures 8 and 9 high sized agglomerates. How it will work in the organism? Have you done DLS measurements in aqueous solution? Moreover, I haven't seen any stability experiments in aqueous solution.

5) For hyperthermia experiments I haven't seen any comparison with other works.

6) Discussion section is low. Authors should enlarge discussion section in the relation to biological application of such nanocomposites.

7) The novelty and limitations of the work have to be mentioned in a better way.

Minor comments

In the text many grammatical errors. I highly recommend reading through the text, and extensive English editing.

Moreover, some other chemical error as Iron oxide tetrahydrate FeCl2·4H2O, and iron oxide hexahydrate FeCl3·6H2O, Calcium nitrate Ca(NO3)2·4H2O, can be found. The compound names are wrong.

Fig. 4b. Is it DLS?

Fig 4a It is very difficult to see the sizes.

For citric acid modified IR COOH group 1395 cm-1 is not visible. Figure 5

Author Response

We would like to thank you for your valuable comments and suggestions. We tried to include them all in our work.

Within the synthesis of the proposed composite, we have found out that depending on the synthesis conditions proposed in the literature, the results correlating the physicochemical properties of our composite with the experimental conditions differed from the literature. Therefore, we have decided to focus on the basic parameters described in the paper to show the effect of experimental conditions during synthesis on the properties of the composite until coating the nanocomposite with a drug. Therefore, in this work, the results being an introduction to the drug delivery studies are presented. Addressing the comments, we have corrected as manuscript as suggested.

1) The title of the paper is broad. I recommend clarifying the title of the article.

Indeed, the title was too broad. We propose: Influence on experimental conditions during synthesis on the physicochemical properties of the nanocomposite based on SPIONs and hydroxyapatite for magnetic hyperthermia application

2) Much more 2-3 year review papers have to be used for Introduction section to show the perspectives of the area.

Thank you for this suggestion, we have modified references.

3) The novelty of the spions synthesis has to be mentioned. I have seen many papers about the synthesis using NH3 or NaOH (the base differences papers), and pH values, which shows differences in sizes, magnetic properties, etc. Is the synthesis section 2.2 and its properties (section 3.1) are new?

Indeed, the conditions proposed in this work are not new. However, depending on the experimental conditions during synthesis, the material in the nanoscale changes its properties. Therefore, basic studies of the physicochemical properties of the composite were needed to determine the effect of several parameters. As literature presents various parameters, we tried to propose these that give the highest magnetization saturation of the core. Therefore, based on these conditions, the following coating was performed.

4) For the biological applications the sizes of the nanocomposites must be between 15-200 nm. However, I see SEM and TEM figures 8 and 9 high sized agglomerates. How it will work in the organism? Have you done DLS measurements in aqueous solution? Moreover, I haven't seen any stability experiments in aqueous solution.

The agglomerates presented in the SEM and TEM images are caused by the sample drying. We present the first stage of the studies, where we propose the optimal composition of the nanocomposite. The stabilization of the suspension will be needed for further investigations of the drug loading to get a stable suspension. In this work, we focused on the magnetic core-porous coat.

5) For hyperthermia experiments I haven't seen any comparison with other works.

We have added the comparison of results with other works.

6) Discussion section is low. Authors should enlarge discussion section in the relation to biological application of such nanocomposites.

Thank you for this suggestion. We have modified the discussion and conclusions section.

7) The novelty and limitations of the work have to be mentioned in a better way.

Novelty and limitations are included in the corrected version of the manuscript.

Minor comments

In the text many grammatical errors. I highly recommend reading through the text, and extensive English editing.

Moreover, some other chemical error as Iron oxide tetrahydrate FeCl2·4H2O, and iron oxide hexahydrate FeCl3·6H2O, Calcium nitrate Ca(NO3)2·4H2O, can be found. The compound names are wrong.

Indeed, we missed it. The text is corrected in the text.

Fig. 4b. Is it DLS?

No, the presented histogram is based on the TEM images, where the data were analyzed within the ImageJ software. We have modified the manuscript by adding the DLS analysis and mentioned the TEM-based histogram as well.

Fig 4a It is very difficult to see the sizes.

The image was analyzed within the ImageJ software, where the results were compared with different spots on the sample.

For citric acid modified IR COOH group 1395 cm-1 is not visible. Figure 5

The intensity of this peak is very low for the low content of the citrates onto the SPIONs surface (of about a few % by mass max.). However, such an amount is sufficient for the stabilization of SPIONs for the following HAp coating.

Reviewer 3 Report

The authors fabricated magnetic nanocomposites consisting of superparamagnetic iron oxide nanoparticles and hydroxyapatite layers with possible application in magnetic hyperthermia. The synthesis of iron oxide nanoparticles by coprecipitation methods is already well reported. Also, the real structure of nanocomposites is not confirmed very well by instrumental techniques such as low-resolution TEM images. The novelty of this work is very low. However, the revised manuscript should be considered for publication. The following comments must be addressed before publication.

1) The introduction section should be rewritten with the specific aims of the current research. The appropriate reference (CA Cancer J Clin . 2021, 71(3):209-249) must be cited for the statement of cancer statistics. Additionally, authors should read/cite the following reference for revising the introduction. i) Journal of Colloid and Interface Science 579 (2020) 186–194. ii) Cancers 2021, 13, 2735. iii) Colloids and Surfaces B: Biointerfaces 174 (2019) 42–55.

2) DLS-based size distribution measurement of nanoparticles and nanocomposites over time should be included and explain their colloidal stability.

3) The TEM images demonstrate the significant aggregation of nanoparticles. What is the effect of nanoparticle aggregation on heating efficiency during magnetic hyperthermia measurement?

4) Are the applied magnetic fields under the biological safe regions? If so, what are the SAR values of nanocomposites?

Author Response

We would like to thank you for your valuable comments and suggestions. We tried to include them all in our work.

Within the synthesis of the proposed composite, we have found out that depending on the synthesis conditions proposed in the literature, the results correlating the physicochemical properties of our composite with the experimental conditions differed from the literature. Therefore, we have decided to focus on the basic parameters described in the paper to show the effect of experimental conditions during synthesis on the properties of the composite.

Addressing the comments, we have corrected as manuscript as suggested.

1) The introduction section should be rewritten with the specific aims of the current research. The appropriate reference (CA Cancer J Clin . 2021, 71(3):209-249) must be cited for the statement of cancer statistics. Additionally, authors should read/cite the following reference for revising the introduction. i) Journal of Colloid and Interface Science 579 (2020) 186–194. ii) Cancers 2021, 13, 2735. iii) Colloids and Surfaces B: Biointerfaces 174 (2019) 42–55.

Thank you for this suggestion. The proposed papers are very interesting in delivering valuable information, so we have included them in the citations.

2) DLS-based size distribution measurement of nanoparticles and nanocomposites over time should be included and explain their colloidal stability.

We have added the hydrodynamic size distribution for the samples tested within the magnetic hyperthermia.

3) The TEM images demonstrate the significant aggregation of nanoparticles. What is the effect of nanoparticle aggregation on heating efficiency during magnetic hyperthermia measurement?

In this work, we tested several samples in many conditions, where each sample was prepared min. Three times and compared with other samples to deliver the data that are repeatable. Therefore, we have focused on the TEM studies. For the number of samples and equipment access limitations, we did not measure HR-TEM.

4) Are the applied magnetic fields under the biological safe regions? If so, what are the SAR values of nanocomposites?

Yes, the magnetic fields were applied under the biological safe regions, an order of magnitude below the Brezovich limit. Suggestions are included in the text.

Round 2

Reviewer 2 Report

The author revised the manuscript. The paper may be accepted in present form.

Reviewer 3 Report

The author revised the manuscript. It is now ready for publication.